# Designing lithium halide solid electrolytes

Qidi Wang ®[1], Yunan Zhou[2], Xuelong Wang[3], Hao Guo[4], Shuiping Gong[5], Zhenpeng Yao ®[5], Fangting Wu[2], Jianlin Wang ®[6], Swapna Ganapathy ®[1], Xuedong Bai ®[6], Baohua Li ®[2], Chenglong Zhao ®[1]✉, Jürgen Janek[7]✉ & Marnix Wagemaker ®[1]✉

All-solid-state lithium batteries have attracted widespread attention for next-generation energy storage, potentially providing enhanced safety and cycling stability. The performance of such batteries relies on solid electrolyte materials; hence many structures/phases are being investigated with increasing compositional complexity. Among the various solid electrolytes, lithium halides show promising ionic conductivity and cathode compatibility, however, there are no effective guidelines when moving toward complex compositions that go beyond *ab*-initio modeling. Here, we show that ionic potential, the ratio of charge number and ion radius, can effectively capture the key interactions within halide materials, making it possible to guide the design of the representative crystal structures. This is demonstrated by the preparation of a family of complex layered halides that combine an enhanced conductivity with a favorable isometric morphology, induced by the high configurational entropy. This work provides insights into the characteristics of complex halide phases and presents a methodology for designing solid materials.

All-solid-state batteries (ASSBs), using inorganic solid electrolytes (SEs), are promising to meet the growing demands on energy storage systems, potentially providing higher energy density and safety over commercial liquid electrolyte lithium (Li)-ion batteries[1]. Replacing liquid electrolytes with SEs can resolve intrinsic drawbacks, such as low Li-ion transference number, complex solid-liquid interface reactions, and thermal stability, which extends battery cycle life. Despite the impressive progress in identifying SEs and establishing structure-property relationships, many challenges remain toward application in ASSBs, motivating fundamental research[2–7].

Li-containing halide compounds with the general formula, $Li_{3+m}Me_{1+n}X_6$, are considered as a promising family of electrolyte materials for ASSBs owing to their relatively high room-temperature conductivity and good compatibility with high-specific-capacity oxide cathodes[6–11], making halide SEs potentially attractive for high-power and high-energy density batteries. Here, Me stands for one or multiple metal elements and X for one or multiple halogen elements. The development of halides has largely proceeded through extending known compounds into new compositions, usually guided by ionic radii ($R$) of the ions[12,13]. However, when moving towards increased compositional complexity, it is desirable to further develop guidelines for the design of halide SEs. In this context, high-entropy (HE) is relevant, which has been brought forward by introducing five or more principal elements in alloys[14,15]. Increasing the number of principle elements increases configurational entropy, changing both thermodynamic stability as well as kinetic properties of the materials, specifically for SEs showing potential advantages[16]. As compositional space increases rapidly with the number of principle elements, the design of

[1]Department of Radiation Science and Technology, Delft University of Technology, Delft 2629JB, the Netherlands. [2]Shenzhen Key Laboratory on Power Battery Safety and Shenzhen Geim Graphene Center, School of Shenzhen International Graduate, Tsinghua University, Guangdong 518055, China. [3]Chemistry Division, Brookhaven National Laboratory, New York 11973, USA. [4]Neutron Scattering Laboratory, Department of Nuclear Physics, China Institute of Atomic Energy, Beijing 102413, China. [5]The State Key Laboratory of Metal Matrix Composites, School of Materials Science and Engineering, Center of Hydrogen Science, Innovation Center for Future Materials, Zhangjiang Institute for Advanced Study, Shanghai Jiao Tong University, Shanghai 200240, China. [6]State Key Laboratory for Surface Physics, Institute of Physics, Chinese Academy of Sciences, Beijing 100190, China. [7]Institute of Physical Chemistry, Center for Materials Research, Justus-Liebig-University Giessen, Giessen D-35392, Germany. ✉e-mail: c.zhao-1@tudelft.nl; juergen.janek@pc.jlug.de; m.wagemaker@tudelft.nl

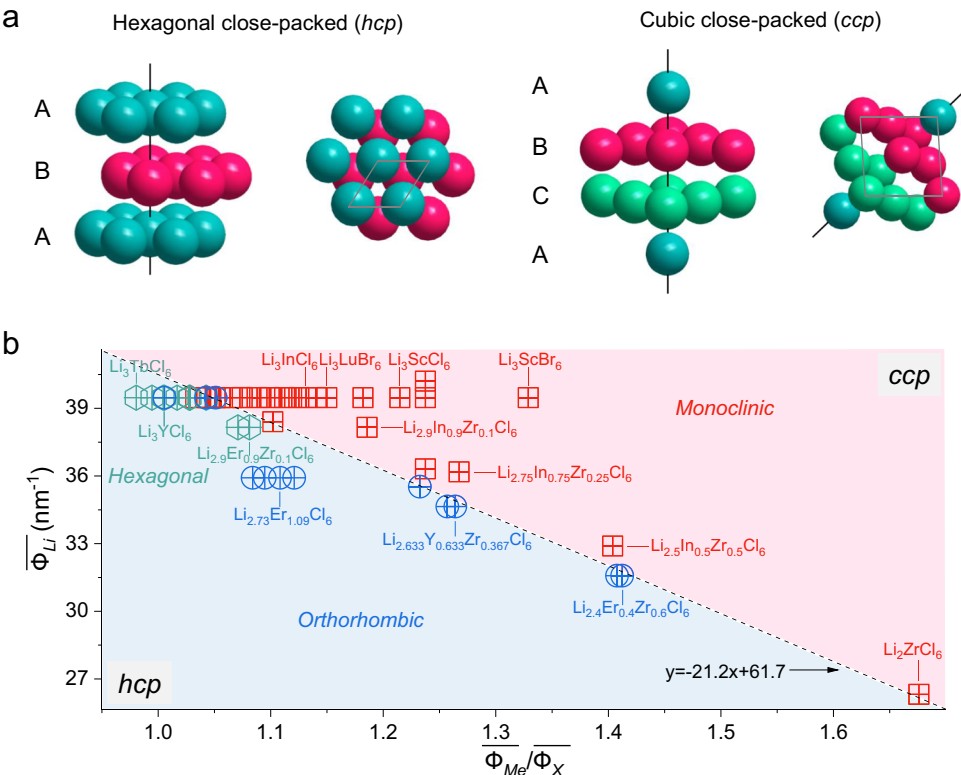

**Fig. 1 | Ionic potential and its use in Li-ion halides. a** Schematic illustration of halide close-packed structures in the hexagonal close-packed (*hcp*), e.g., hexagonal (*P-3m*1) and orthorhombic (*Pnma*) arrangements, and the cubic close-packed (*ccp*), e.g., monoclinic (*C2/m*) arrangements. **b** Phase map showing representative halide materials as a function of ionic potential ratio and Li-ion potential, accounting for Li content, oxidation state, and composition of different Me ions (Supplementary Tables 5–9). The phase map indicates a distinction between *hcp* and *ccp* X-ion sublattices. Notably, the hexagonal (hexagon) and orthorhombic (circle) structures both have a *hcp* X-ion sublattice, which is suggested to cause partial overlap. The dividing lines are calculated with current reference to the boundaries of *hcp* and *ccp* halides.

complex halides, will benefit from the development of relationships between the composition, structure, and functional properties. Halides can form different compositions, $Li_2MeX_4$, $Li_3MeX_6$, $Li_2MeX_6$, or $LiMeX_6$, as the charge number of the Me ions increases from 2 to 5, respectively (Supplementary Figs. 1–15, Table 1 and note 1). This large compositional diversity provides ample opportunities to tune properties toward the demands of ASSBs. However, the diverse structural chemistry makes it difficult to effectively design a specific structure based on possible compositions. This is also difficult to assess with computational approaches such as density functional theory (DFT) simulations due to the vastly increasing configurational space with huge compositional complexity.

Among the various structures of halide SEs, monoclinic (*C2/m*) structures, hexagonal (*P-3m*1), and orthorhombic (*Pnma*) have attracted significant attention because they offer relatively high ionic conductivities[6,7,12,17]. For the hexagonal and orthorhombic structures, the X sublattice has a hexagonal close-packed (*hcp*) stacking, which includes compositions such as $Li_3MeCl_6$ (Me=Tb-Yb) and $Li_3MeCl_6$ (Me=Y, Yb and Lu). In the monoclinic structure, the X sublattice has the cubic close-packed (*ccp*) stacking, which includes compounds such as $Li_3MeCl_6$ (Me=Sc and In) and $Li_3MeBr_6$ (Me=Y, In and Sm-Lu) (Fig. 1a and Supplementary Fig. 2). Based on the close-packed crystals, the ratio of the metal to the anion radius, $\frac{R_{Me}}{R_X}$, has been proposed to relate the phase stability for given compositions[12], where the Me–X coordination evolves from [MeX$_3$] to [MeX$_8$] as the ratio increases from 0.155 to 1.0. However, this ratio of radii only applies to the [MeX$_6$] configuration in the range of $\frac{R_{Me}}{R_X}$ = 0.414–0.732, which is unable to distinguish *ccp* monoclinic (*C2/m*) from the *hcp* hexagonal (*P-3m*1) and orthorhombic (*Pnma*) structures for known and relatively simple compositions (Supplementary Fig. 16 and Tables 2–4). This is understandable because the ionic radii do not account for the contribution of the

charge of the metal ions, which determines the strength of ionic interactions, influencing structure stability. Especially, when the composition becomes complex with more than one metal, having different charge numbers, and/or halogen elements, ionic radii fall short as descriptors towards the design and discovery of new halide materials, as demonstrated by Supplementary Fig. 16.

The goal of the present work is to develop an effective and practical guideline for the design of complex halide materials, aiming to identify halides with improved properties in a specific crystal structure. We show that ionic potential, a methodology of quantifying interactions in inorganic crystalline compounds, can capture the structural difference of layered halide materials, making it possible to distinguish and design phases with tunable compositions. This is demonstrated by the prediction and preparation of a family of HE halide SEs that possess high (more isotropic) ionic conductivity (>2.0 mS cm$^{-1}$) as well as an isometric morphology. Morphology, especially, is an important practical asset that facilitates the densification of the SEs during electrolyte processing. The ability to predict the stacking based on composition alone, as well as inducing favorable isotropic properties through compositional complexity, provides promising developing directions toward designing Li-containing halide electrolytes for future ASSBs.

## Results

### Ionic potential as a phase descriptor for Li halides

Crystal lattices of halides can be represented by ionic Me–X bonds in the close-packed anion sublattices. The nature of Me and X ions is decisive for the hybridization of the Me–X bonds, which determines the structure and thus impacts Li-ion diffusivity in the different structures. The ionic bonds in halides present a very different situation to most other SEs having more covalent P–O/S or B–H bonds[5,18], where

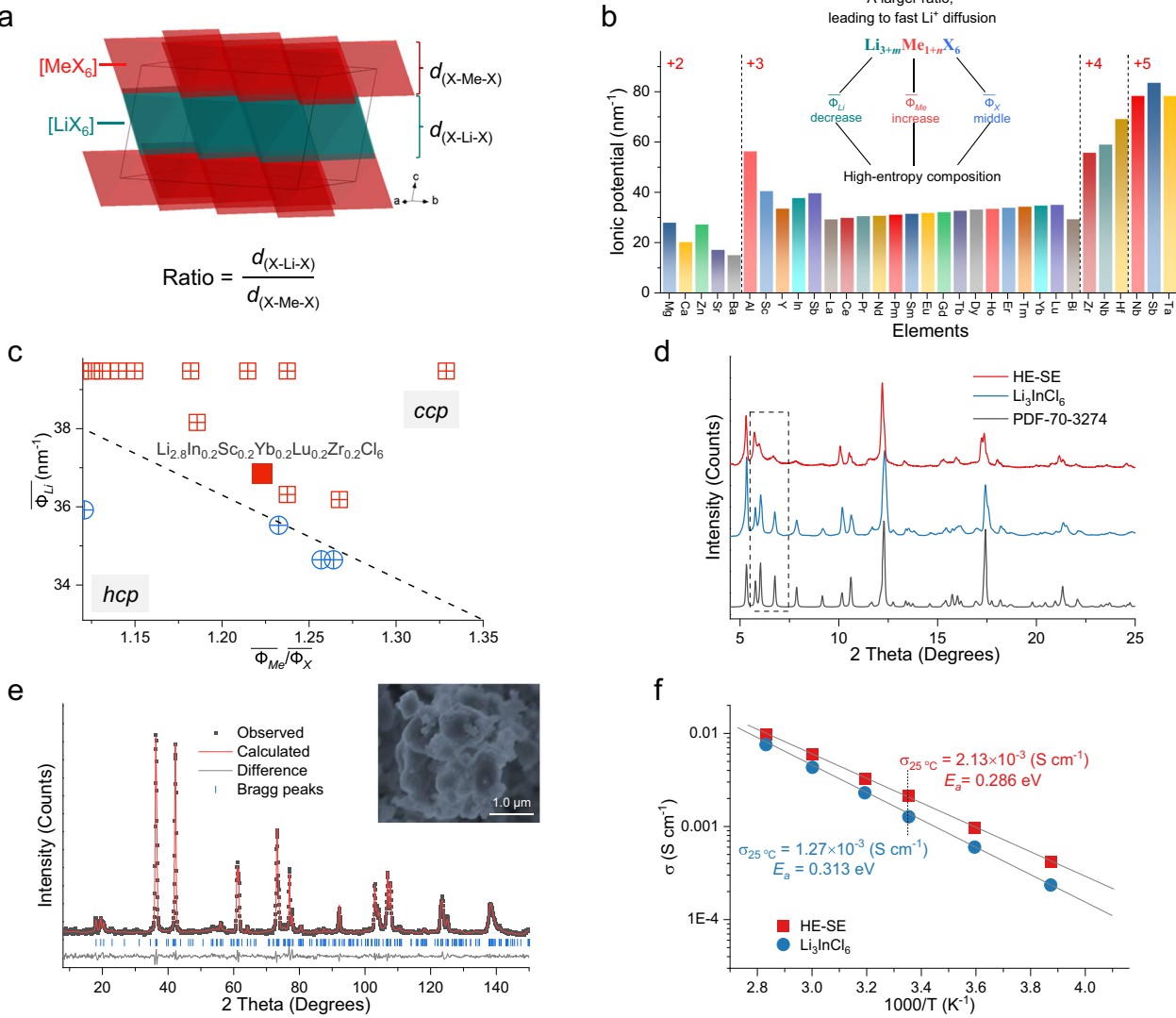

**Fig. 2 | Designing layered Li-ion halide SEs. a** Crystal structure of layered Li-ion halides. The interlayer distance $d_{(X-Li-X)}$ is the average perpendicular distance between the two halogen sheets enclosing Li ions, and the interlayer distance $d_{(X-Me-X)}$ is the perpendicular distance of two parallel sheets containing Me ions. The ratio is determined by the interlayer distances of $d_{(X-Li-X)}$ and $d_{(X-Me-X)}$ for the layered halides. **b** Ionic potential of possible cations used in Li-ion halide SEs including Me ions with various oxidation states. The inset shows the principles for the fast Li-ion diffusion layered halide SEs, referred to as the structural characteristics. **c** Designing HE layered halide SE guided by the ionic potential method (Supplementary Table 10). **d** XRD patterns of the targeted HE-SE $Li_{2.8}In_{0.2}Sc_{0.2}Yb_{0.2}Lu_{0.2}Zr_{0.2}Cl_6$ material, the $Li_3InCl_6$ and the standard $C2/m$ reference pattern. **e** NPD pattern of HE-SE $Li_{2.8}In_{0.2}Sc_{0.2}Yb_{0.2}Lu_{0.2}Zr_{0.2}Cl_6$ material (Supplementary Tables 11 and 12). The inset shows the SEM image of the morphology. **f** Li-ion conductivity and activation energy of SEs.

halides are more similar to the well-established layered transition-metal (TM) oxides with the general composition $Li_xTMO_2$[19] used as cathodes in batteries. For example, the layered halide $Li_3InCl_6$ SE and the layered $Li_2RuO_3$ cathode share the same monoclinic $C2/m$ structure[20,21], suggesting the importance of considering the charge of the ions. Here we investigate the effectiveness of ionic potential as a phase descriptor for Li-ion halide materials. The ionic potential is defined as $\Phi = \frac{n}{R}$, where $n$ is the dimension-less ion charge number and $R$ the ion radius, thus reflecting the surface charge density of an ion[22]. By utilizing the ionic potential method, not only the impact of the ion radii, but also the charge density is included, which is important for ionic crystals. This is brought forward as a simple descriptor that effectively represents the ionic interactions at play in this family of halide structures. Hence, the effective ionic potential ratio of $\frac{\overline{\Phi_{Me}}}{\overline{\Phi_X}}$ between Me and X ions is proposed, which describes the effective Me−X hybridization interaction. $\overline{\Phi_{Me}}$ represents the weighted average

ionic potential of Me ions, defined as $\overline{\Phi_{Me}} = \sum \frac{w_i n_i}{R_i}$, where $w_i$ is the fraction of metal element $Me_i$ having charge number $n_i$ and radius $R_i$. $\overline{\Phi_X}$ is the weighted average ionic potential of halogen X defined as $\overline{\Phi_X} = \frac{w_j}{R_j}$, where $w_j$ is the fraction of halogen element $X_j$ with ion radius $R_j$, having an absolute charge number of 1. The X could also be the anion of oxygen with the absolute charge number of 2, or other anions. Assuming a general formula $Li_yMe_zX_6$, $\overline{\Phi_{Li}}$ represents the weighted average ionic potential of Li defined as $\overline{\Phi_{Li}} = \frac{y}{R_{Li}}$, and charge neutrality of $Li_yMe_zX_6$ demands $\sum w_i n_i + y = 6$.

Taking into account the Li-ion potential $\overline{\Phi_{Li}}$, representing the Li composition, is critical for the electrostatic interactions within the halide structures, where plotting $\overline{\Phi_{Li}}$ versus $\frac{\overline{\Phi_{Me}}}{\overline{\Phi_X}}$ for reported compositions presents a phase map as shown in Fig. 1b. The relatively distinct regions of the different structures suggest that the ionic potential is an

effective descriptor of the hybridization interaction of the Me–X bonds, reflecting the electrostatic repulsion between [LiX$_6$] and [MeX$_6$] octahedra in the three different lattices within two close-packed configurations. Generally, at a fixed Li content, a larger ionic potential of Me indicates a larger surface charge, resulting in a larger electrostatic repulsion between [LiX$_6$] and the adjacent [MeX$_6$] octahedra. This promotes the formation of the monoclinic (C2/m) structure having a ccp halogen sublattice arrangement. In contrast, a relatively small ionic potential of Me ions, weakens this electrostatic repulsion, favouring the hcp arrangement of hexagonal (P-3m1) or orthorhombic (Pnma) lattices. Interestingly, the larger electrostatic repulsion between the [MeX$_6$] and [LiX$_6$] octahedra, as present in the monoclinic (C2/m) structure, typically results in higher conductivities with values reaching $10^{-3}$ S cm$^{-1}$ at room temperature, whereas the other two structures result in lower conductivities in the range of ~$10^{-5}$-$10^{-4}$ S cm$^{-1}$ [6,12,23]. This finding demonstrates that the charge of ions plays an important role in the halide structures, which is not sufficiently incorporated through the ion radius alone. This motivates the formulation of the effective ionic potential ratio vs. the Li-ion potential as an accurate descriptor for the design of complex Li-ion halide SEs, which are very challenging to assess with quantum chemistry methods, like DFT. In addition, as for the recently reported spinel Li$_2$(Sc/In)$_{2/3}$Cl$_4$[17], which can be expressed as Li$_3$(Sc/In)Cl$_6$, the ionic potential method correctly predicts the monoclinic ccp structure.

**Designing layered Li-ion halide HE-SEs**

Halides having the monoclinic (C2/m) structure, and thus a ccp halogen stacking, generally show a relatively high ionic conductivity compared to the other two halide structures. In the layered structure, MeX$_2$ slabs are alternated by LiX$_2$ slabs consisting of [MeX$_6$] and [LiX$_6$] octahedra, respectively. The electrostatic cohesive and repulsive interactions between the slabs can be reflected by the ratio between the interlayer distances $d_{(X–Li–X)}$ and $d_{(X–Me–X)}$ (Fig. 2a), affecting the Li-ion diffusion. The interlayer distance $d_{(X–Li–X)}$ is the average perpendicular distance between the two halogen sheets enclosing the Li ions, and the interlayer distance $d_{(X–Me–X)}$ is the perpendicular distance of two parallel sheets containing the Me ions. To a certain extent, a larger ratio of these two distances, $d_{(X–Li–X)}/d_{(X–Me–X)}$, can result in a higher Li-ion diffusivity, owing to the relatively larger Li-ion layer width[24]. Hereby ionic potential can be used to improve the Li-ion conductivity (Fig. 2b and Supplementary Tables 5 and 6). The interlayer distance $d_{(X–Li–X)}$ can be increased by lowering the average Li-ion potential $\overline{\Phi_{Li}}$, equivalent to lowering the Li-ion content, or alternatively by increasing the Me–X bond strength, which can be realized by introducing Me ions with larger ionic potentials. The halogen X functions as a link between adjacent Me and Li ions, which balances the relative interaction between the halogen and Me and Li, respectively. The effective ionic potential ratio can be increased within a specific range through properly tuning the ionic potential of Me and X, which can weaken the Li–X interaction, thus improving the Li-ion mobility and conductivity. However, it is important to note that increasing the effective ionic potential ratio above a certain threshold can have the opposite effect. For most cases, the Me ions used in the halides with a higher charge number generally exhibit relatively smaller ionic radii, as demonstrated in Supplementary Table 5. For example, the ionic radius of Zr$^{4+}$ (in 6-coordination) is 0.072 nm, which is smaller than that of Sc$^{3+}$ (0.0745 nm), Y$^{3+}$ (0.09 nm) and In$^{3+}$ (0.08 nm). Consequently, the introduction of more Me ions with higher charge numbers results in an increased ionic potential, $\overline{\Phi_{Me}}$, leading to a decrease in $d_{(X–Me–X)}$ and increase in $d_{(X–Li–X)}$. This does not necessarily increase Li-ion conductivity, for example the monoclinic Li$_2$ZrCl$_6$[25] exhibits an ion conductivity of approximately $10^{-6}$ S cm$^{-1}$ at room temperature, which is lower than that of Li$_3$InCl$_6$. This anomaly can be attributed to the reduced number of charge carriers, resulting in lower ionic conductivity[26,27]. Thus, optimization of the number of charge carriers

and vacancies is additionally required to achieve maximum conductivity.

Using the ionic potential as a guide, a complex halide with five Me species is proposed aiming to have the monoclinic structure and also aiming at tuning the effective ionic potential ratio to realize a high Li-ion conductivity. Additional criteria that we set are: (1) Use Cl as halogen to provide a high oxidation stability[28]. (2) A Li composition with 2.8 mol fraction (in the Li$_3$MeX$_6$ formula unit) to balance the number of charge carriers/vacancies (achieved by charge compensation in the Me ions). This is motivated by previously reported SEs that were optimized towards Li-ion conductivity, for example, Li$_{6.5}$La$_3$Zr$_{1.5}$Ta$_{0.5}$O$_{12}$ derived from the parent composition Li$_7$La$_3$Zr$_2$O$_{12}$. (3) An equal proportion of five Me ions to maximize the configurational entropy. Considering the definition of the ionic potential and the phase map shown in Fig. 1b, the average Li-ion potential will be 36.84 nm$^{-1}$, hence to fall within the monoclinic phase space requires an effective ionic potential ratio higher than 1.18. These criteria can for instance be met by Li$_{2.8}$In$_{0.2}$Sc$_{0.2}$Yb$_{0.2}$Lu$_{0.2}$Zr$_{0.2}$Cl$_6$ (all Me ions with a relatively large ionic potential in Fig. 2b), resulting in a ratio of 1.22 for this as-proposed HE halide SE to have the monoclinic layered structure (Fig. 2c, Supplementary Table 10).

Li$_{2.8}$In$_{0.2}$Sc$_{0.2}$Yb$_{0.2}$Lu$_{0.2}$Zr$_{0.2}$Cl$_6$ (HE-SE) was prepared by a typical solid-state reaction. X-ray diffraction (XRD) and Neutron powder diffraction (NPD) confirm the layered, monoclinic structure (Fig. 2d, e), consistent with the ionic potential prediction. The pure phase implies that the multiple Me cations occupy in same lattice sites with an increased disorder. The well-studied Li$_3$InCl$_6$ is prepared as a reference for the representative layered compound (Fig. 2d). Comparing the XRD patterns of both materials, the broadened superstructure peaks between 5.5° to 10° indicate that the HE-SE has a higher concentration of stacking faults, presumably due to the multiple element composition. The prepared HE-SE has a larger ratio of interlayer distances ($d_{(X–Li–X)}/d_{(X–Me–X)} = 1.119$) compared to that of Li$_3$InCl$_6$ (1.097) (Supplementary Fig. 17), reflecting the larger Li-ion layer distance. The combined Rietveld refinement of NPD and XRD data of the HE-SE results in a partially disordered arrangement of Li and Me ions in the MeX$_2$ slabs, with an increased fraction of the Li residing in the Me (4 g) site (Supplementary Tables 11 and 12), which results in more vacant Li-ion positions in the LiX$_2$ slabs. In addition, compared to the Li$_3$InCl$_6$ material prepared with the same method, the HE-SE shows smaller and more spherical particles (Fig. 2e, Supplementary Figs. 18 and 19). Transmission electron microscope (TEM) images and energy dispersive X-ray spectroscopy (EDS) maps show that all Me elements are uniformly distributed with increased disorder (Supplementary Fig. 20). Solid-state $^6$Li magic angle spinning nuclear magnetic resonance (MAS-NMR) measurements are employed to study the local Li-ion environment (Supplementary Fig. 21). The HE-SE results in a considerably broader resonance (Full width at half maximum, FWHM, approximately 1.65 ppm) compared to that of Li$_3$InCl$_6$ (FWHM approximately 0.1 ppm) (Supplementary Fig. 21a), which is more obvious under lower temperature (Supplementary Fig. 21b), suggesting a disordered distribution in Li-environments. This can be the result from the complex composition in conjunction with a change in Li-ion site occupancies as observed from the diffractions.

Li-ion conductivity of the HE-SE was examined by temperature-dependent electrochemical impedance spectroscopy (EIS) measurements at various temperatures (Fig. 2f, Supplementary Fig. 22). The as-prepared HE-SE exhibits a higher ionic conductivity of around 2.13 mS cm$^{-1}$ at 25 °C (the sum of the grain boundary and bulk resistances) and lower activation energy of around 0.286 eV compared to the 1.27 mS cm$^{-1}$ and ~0.313 eV for the Li$_3$InCl$_6$ (Fig. 2f), where the as-prepared Li$_3$InCl$_6$ is in agreement with the reported results using the solid-state synthesis[29]. The electronic conductivity of the HE-SE was measured using direct current (DC) polarization at voltages from 0.2 up to 1.0 V at room temperature, resulting in a lower electronic

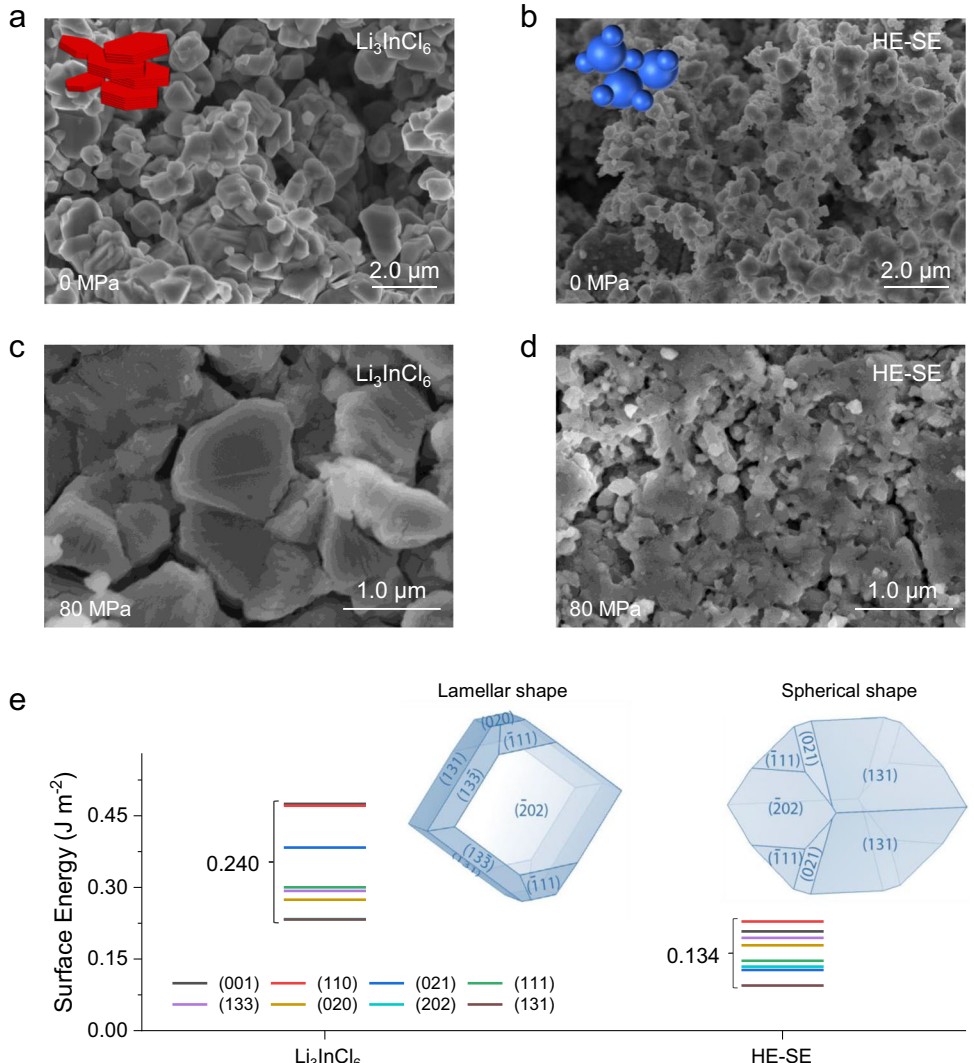

**Fig. 3 | Morphology of layered halide SEs. a, b** SEM images of $Li_3InCl_6$ and HE SEs. Insets illustrate the corresponding particle morphologies. **c, d** SEM top view of SE pellets after compression. **e** Surface energies for typical crystal faces of $Li_3InCl_6$ and HE SEs. Insets show Wulff shapes of the SEs.

conductivity of $3.3 \times 10^{-9}\,S\,cm^{-1}$ compared to the $Li_3InCl_6$ reference having a value of $6.2 \times 10^{-9}\,S\,cm^{-1}$ (Supplementary Figs. 23 and 24). The bulk microscopic Li-ion diffusion mechanism was studied using solid-state $^7Li$ NMR by measuring the temperature-dependent $^7Li$ static spin-lattice relaxation (SLR) rate using a saturation recovery experiment[30–32] (Supplementary Figs. 25–27 and Supplementary note 2). As shown in Supplementary Fig. 25, the temperature-dependent spin-lattice relaxation rate of $Li_3InCl_6$ reflects two distinguishable peaks, suggesting the presence of two Li-ion jump processes that differ in their average jump frequency. Considering the two-dimensional layered structure, this has been suggested to be the result of the relatively fast intralayer diffusion within the $LiCl_2$ slabs and relatively sluggish diffusion across the $InCl_2$ slabs, thus between two $LiCl_2$ slabs[33]. In contrast, a symmetric spin-lattice relaxation rate is observed for HE-SE material, suggesting that Li-ion diffusion within the $LiCl_2$ slabs and diffusion across the $MeCl_2$ slabs between two $LiCl_2$ slabs have a similar average jump frequency, which suggests a more homogeneous jump process[30,32,34]. In combination with the observed stacking faults and highly disordered distribution of Li (Fig. 2d), as well as the increased cation disorder resulting from the presence of multiple Me ions, this indicates more three-dimensional diffusion in the HE-SE compared to $Li_3InCl_6$, which is suggested to be the origin of the improved conductivity[33,35].

## Morphology characteristics of HE layered materials

The complex composition of the HE-SEs not only impacts the crystal structure but is also found to impact the morphology. Generally, layered halide SEs have an anisometric morphology, often forming agglomerates, such as the reported for $Li_3InCl_6$[12], $Li_3ScCl_6$[36], and $Li_3YBr_6$. While $Li_3InCl_6$ and HE-SE were prepared using the same synthesis method, they show a very different morphology (Fig. 3a, b). $Li_3InCl_6$ has the typical lamellar stacking morphology consistent with previous observations[12], whereas the layered HE-SE exhibits smaller, more spherical particles having a relatively smooth surface (Fig. 3b). The electrolyte particle morphology affects grain boundary formation, and thus plays an important role in the long-range Li-ion transport in ASSBs[37]. Because pressure can affect the grain boundary resistance[38], the pressure-dependent impedance of the prepared SEs was acquired (Supplementary Fig. 28). The relative impedance as the function of pressure indicates that the spherical morphology of the HE-SE leads to a smaller drop. This implies a denser packing of the HE-SE, which is confirmed by comparing SEM images of both SE pellets (Fig. 3c, d and Supplementary Figs. 29 and 30).

Since the surface energy is decisive in the formation of the particle morphology[39], the surface energies of several distinct crystal facets of $Li_3InCl_6$ and HE-SE materials were calculated to gain qualitative insights. It should be noted that the disorder of these systems cannot

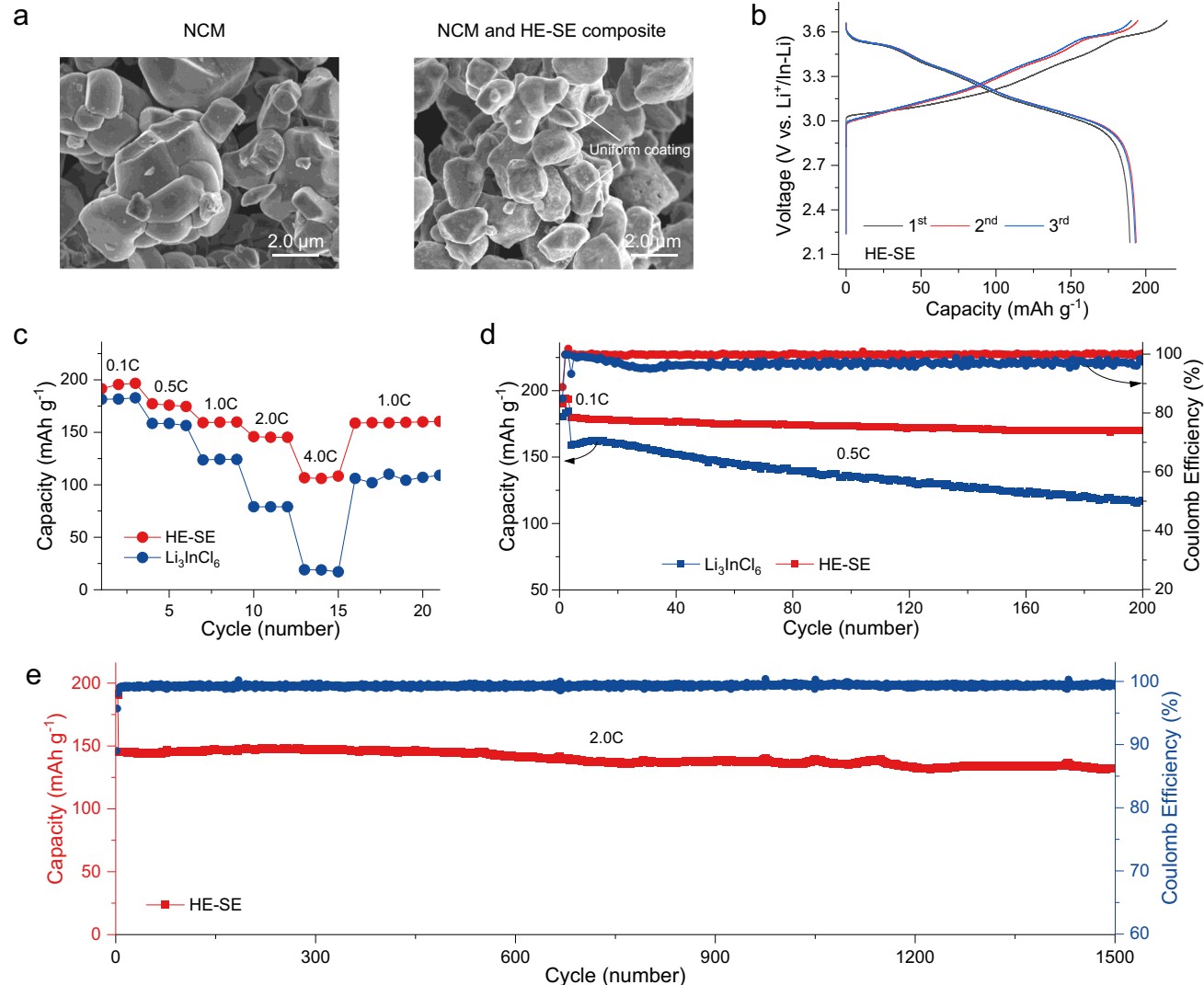

**Fig. 4 | Electrochemical properties of ASSBs. a** Morphologies of NCM cathode and cathode composite with HE-SE material. Around 10% HE-SE is used for the cathode coating. **b** Galvanostatic charge-discharge curves cycled at a rate of 0.1 C (1 C = 190 mA g$^{-1}$) in the voltage range of 2.18−3.68 V vs. Li$^+$/In-Li. **c** Retention of the discharge capacity during cycling at various rates from 0.1 up to 4.0 C. **d** Discharge capacity retention and Coulomb efficiency upon cycling. Cells were pre-cycled at 0.1 C for the first three cycles to activate the cathode material, followed by cycling at 0.5 C. **e** Discharge capacity retention and Coulomb efficiency upon cycling, the first three cycles at 0.1 C, followed by cycling at 2.0 C.

be accurately captured in the supercells used in the simulations, yet the results provide trends that can be used for the qualitative understanding pursued (Fig. 3e, Supplementary Figs. 31, 32, and note 3). The results for various facets of both materials are shown in Fig. 3e, demonstrating that the distribution in surface energies (0.240 J m$^{-2}$) is wider for Li$_3$InCl$_6$ and that the average value is larger. According to Wulff's law, the wider the surface energy distribution is, the more preferential growth can be expected, exposing the low-energy crystal facets shown in the Wulff shapes in Fig. 3e. Consistently, Li$_3$InCl$_6$ particles grow more anisometric to form elongated crystals. In contrast, the HE-SE material prefers to grow more spherical, more isometric consistent with the SEM observations.

### Electrochemical performance of HE-SEs

We further assessed the application of the as-prepared HE-SE in ASSBs. These batteries consisted of single crystalline LiNi$_{0.84}$Co$_{0.07}$Mn$_{0.09}$O$_2$ (NCM) cathode and an In/InLi anode, where all testing was conducted at room temperature. Firstly, the pristine cathode composites were investigated using SEM and EDS mapping, the results of which reveal that the soft HE-SE material is coated to the surface of the NCM

particles, forming a thin and uniform layer through manual grinding (Fig. 4a and Supplementary Fig. 33). When cycled within a voltage range of 2.18−3.68 V vs. Li$^+$/In-Li at a rate of 0.1 C, the ASSBs exhibited a reversible capacity of over 190 mAh g$^{-1}$, with an initial Coulomb efficiency of around 88% (Fig. 4b). The corresponding $dQ/dV$ curves exhibit the anticipated oxidation-reduction processes of the NCM cathodes (Supplementary Fig. 34). An improved rate performance is demonstrated by cycling the batteries at different current densities (Fig. 4c). The reversible capacities achieved amount approximately 195.6, 175.9, 159.6, 145.3, and 106.2 mAh g$^{-1}$ at rates of 0.1, 0.5, 1.0, 2.0, and 4.0 C, respectively. These values significantly surpass those of the reference material Li$_3$InCl$_6$, which exhibited capacities of approximately 181.2, 158.5, 124.3, 79.1, and 18.9 mAh g$^{-1}$ under the same conditions. Furthermore, after the rate cycling test, a reversible capacity of approximately 159.0 mAh g$^{-1}$ was recovered at 1.0 C. We further investigated the cycling stability of the batteries using the as-prepared HE-SE (Fig. 4d), which shows around 95% capacity retention after 200 cycles at 0.5 C (without considering the formation cycles). This exceeded the 73% capacity retention observed for the Li$_3$InCl$_6$ reference material at 0.5 C. Moreover, the HE-SE demonstrated the longer

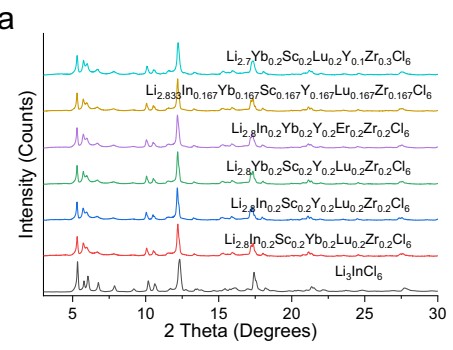

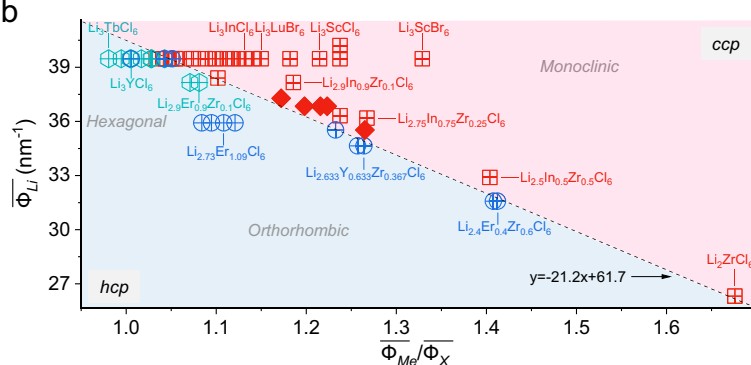

**Fig. 5 | Extension of HE-SEs based on ionic potential method. a** XRD patterns of a family HE-SEs. These new compositions are obtained based on the analysis of ionic potential, considering the different Me ions and Li contents. **b** Phase map based on the ionic potential for Li-ion halides, indicating the as-prepared HE-SEs materials (solid rhombus symbols) having the monoclinic structure (Supplementary Table 13).

cycling life for the ASSBs, with a retention rate of approximately 91% at 2.0 C after 1500 cycles. These results are highly competitive compared to reported rate capabilities and cycling stability of halide materials[6,8–10,12,17,36].

To circumstantiate the generality, we use the proposed ionic potential descriptor to prepare a family of HE halides (Supplementary Table 13), which include the various compositions with different Me ions and Li contents. The calculated ionic potential predicts that each composition has a monoclinic ($C2/m$) structure, confirmed by indexing of the corresponding XRD patterns (Fig. 5a, b). All prepared HE-SE materials show relatively high concentration of stacking faults and broadened diffraction peaks[40], indicating a larger degree of disorder in their lattices (Supplementary Fig. 35). The prepared six-component layered HE halide with the composition $Li_{2.833}In_{0.167}Yb_{0.167}Sc_{0.167}Y_{0.167}Lu_{0.167}Zr_{0.167}Cl_6$ is studied in detail. The Rietveld refinement of the NPD demonstrates it has a similar Li-ion arrangement compared to the above-discussed five-component HE-SE (Supplementary Fig. 36, Tables 14 and 15). SEM image and the EDS mappings further indicate that this six-component HE halide material has an isometric morphology (Supplementary Fig. 37). Li-ion conductivity and electronic conductivity measurements result in ~2.2 mS cm$^{-1}$ and $3.7 \times 10^{-9}$ S cm$^{-1}$ at room temperature (Supplementary Figs. 38 and 39), respectively. These results further demonstrate the effectiveness of ionic potential in the halide materials (Fig. 5a). In addition, the HE-SEs without aliovalent substitution, for example of the $Li_3In_{0.2}Sc_{0.2}Yb_{0.2}Lu_{0.2}Y_{0.2}Cl_6$ composition has also been synthesized and studied. As prepared and tested under the same condition, it has an ionic conductivity up to about 1.76 mS cm$^{-1}$, showing only a slightly lower conductivity compared to the materials with aliovalent substitution.

In summary, the ionic potential is presented as an effective descriptor for the hexagonal, orthorhombic and monoclinic halide structures, especially in predicting the *hcp* vs. *ccp* closed packing of Li-ion halides, capturing the electrostatic repulsion between [LiX$_6$] and [MeX$_6$] octahedra. The ionic potential is intimately linked to the polarization power, which serves as a valuable tool for understanding the hybridization interactions within Li/Me−X bonds in halide compounds. This relationship allows us to differentiate between various structures and establish it as an important descriptor in the design of complex compositions, which is particularly advantageous for intricate systems that pose challenges for ab-initio methods, primarily due to their expansive configurational space. Our study suggests that increasing complexity of multi-component halide compositions can provide a route towards improved physical-chemical properties, such as the observed disordered Li-ion distribution that leads to higher ion diffusivity/conductivity and a spherical, isometric particle morphology/shape

(Supplementary Fig. 40). The latter facilitates higher compaction densities, which promotes the overall conductivity of SEs in ASSBs. The observation that the high configurational entropy stabilizes a spherical morphology that is normally incommensurate with the layered lattice structure, suggests important methods of designing materials with specific functional benefits. It is worth noting that the ionic potential has the ability to capture the key interactions of these structures, but as a straightforward calculation based on the composition, it does not account for the dynamic processes during material synthesis. This can be the cases for amorphous phases or metastable structures resulting from mechanical milling or prepared under particular conditions. For example, in addition to the stable phase of ($P$-$3m1$) $Li_3YCl_6$ and ($C2/m$) $Li_2ZrCl_6$, the metastable structures of ($Pnma$) $Li_3YCl_6$[41], and ($P$-$3m1$) $Li_2ZrCl_6$[42] also exist and can be directly prepared by mechanical milling. Here the ionic potential method does not provide a sensible guideline, because one composition has only one calculated value. Overall, the ionic potential serves as a simple descriptor that can help to support the design of complex materials, which is highly relevant for the emerging field of halide materials and other inorganic crystalline materials.

## Methods

### Materials
The halide SEs were prepared by ball milling and subsequent solid-state reaction. Raw materials of LiCl (99.9%), InCl$_3$ (99.99%), ScCl$_3$ (99.99%,), YCl$_3$ (99.99%), YbCl$_3$ (99.99%), LuCl$_3$ (99.99%) and ZrCl$_4$ (99.9%) were used as starting materials from Sigma-Aldrich. The stoichiometric starting materials were weighed and ball milled at 350 rpm for 12 h sealed in a ZrO$_2$ jar using 18 ZrO$_2$ balls in an Ar-filled glovebox with $p(H_2O)/p < 0.1$ ppm, $p(O_2)/p < 0.1$ ppm. During ball milling, the jar was opened to make the samples homogenous at each 5 h in a glovebox. Then, the resulting mixture was sealed in quartz tubes and annealed at 260 °C for 12 h with a heating rate of 2 °C min$^{-1}$ and cooled naturally to room temperature. After that, the obtained material was directly stored in an Ar-filled glovebox to prevent any moisture exposition.

### Materials characterizations
Morphologies of materials were measured on a cold field scanning electron microscope (SEM, HITACH-SU8010) equipped with energy dispersive X-ray spectroscopy (EDS, IXRF SYSTEM, 550i). The X-ray diffraction analysis (XRD) measurements for electrolytes were performed on a Malvern Panalytic Emperean diffractometer (Almelo) equipped with an Ag radiation source ($\lambda_1 = 0.5594$ nm and $\lambda_2 = 0.5637$ nm) and a GaliPIX3D detector operated in 1D mode. The samples were filled in a 150 mm capillary and sealed inside an Argon filled glovebox.

## Electrochemical measurements

Solid-state batteries were fabricated with the prepared HE-SE in the Ar-filled glovebox. Single crystalline $LiNi_{0.84}Co_{0.07}Mn_{0.09}O_2$ (NCM) was used as cathode material from Guangdong Canrd New Energy Technology Co., Ltd. and an In/InLi anode was used as anode. $Li_{5.5}PS_{4.5}Cl_{1.5}$ was synthesized and prepared by a solid-state reaction with stoichiometric LiCl (99.9%), $P_2S_5$ (99.9%), and $Li_2S$ (99.9%) as starting materials. After ball milling at 150 rpm for 2 h with $ZrO_2$ balls, the precursor was sealed in a quartz tube with Ar and then annealed at 500 °C for 15 h. At first, about 60 mg $Li_{5.5}PS_{4.5}Cl_{1.5}$ powder was put into a poly ether ketone cylinder (10 mm diameter) and followed by cold-pressing with 2 tons of pressure for 1 min, and then 40 mg of as-prepared HE-SE was spread out on the top uniformly and pressed with 2 tons for 1 min. Then around 7–8 mg of the NCM and HE-SE composite cathode, which was mixed with a weight ratio of 8:2 by hand in an agate mortar in an Ar-filled glovebox, was spread on the top of the HE-SE with a pressure of 3 tons for 2 min. At last, the In/InLi anode was put on the other side attached to the $Li_{5.5}PS_{4.5}Cl_{1.5}$ electrolyte. The In/InLi anode was prepared by putting a thin In foil (9 mm diameter) on the top of a Li layer (8 mm diameter), with a weight ratio around 9:1. The galvanostatic charge-discharge measurements were recorded using multi-channel battery testing systems (Land CT2001A or Lanhe G340A) at room temperature (around 25 °C) cycled in a voltage window of 2.8–4.3 V vs. $Li^+$/Li (2.18–3.68 V vs. $Li^+$/In-Li) with a constant pressure of 1 ton for the cells during cycling.

## Ionic conductivity measurements

Ionic conductivities were measured using EIS where solid electrolyte powder is pressed into a 10 mm diameter pellet between two stainless steel rods using a hydraulic press at 3 tons for three minutes in an Ar-filled glovebox. The EIS measurements were collected using a custom-made cell on an Autolab (PGSTAT302N) in the frequency range of 0.1 Hz–1.0 MHz with a potential amplitude of 10 mV at temperatures ranging from −15 to 80 °C. The electronic conductivities at room temperature were measured using direct current polarization measurements with applied potentials of 0.2 V, 0.4 V, 0.6 V, 0.8 V, and 1.0 V for 2 min of the samples. The activation energies of the samples were obtained according to the Arrhenius equation

$$\sigma T = \sigma_0 \exp\left(-\frac{E_a}{k_B T}\right) \tag{1}$$

where $\sigma$, $T$, $\sigma_0$, $E_a$, and $k_B$ represent the ionic conductivity, temperature, pre-exponential factor, activation energy and Boltzmann constant, respectively.

## Neutron powder diffraction (NPD) characterization

NPD data were collected on a high-resolution power diffractometer at the China Advanced Research Reactor (CARR) at the China Institute of Atomic Energy. The wavelength was 1.886 Å with a scanning step of 0.075° measured at ~300 K. The sample of about 2 g was loaded into a vanadium can in an Ar-filled glove box. Rietveld refinement was carried out using the GSAS-II software[43].

## Transmission electron microscopy (TEM) characterization

TEM experiments were performed with a scanning transmission electron microscope (STEM) (JEM-ARM300F, JEOL Ltd.), and EDS mappings were recorded at 300 kV with a cold field emission gun and double Cs correctors. The microscope was equipped with Gatan OneView and K2 cameras for image recording. Analysis of the spectra has been performed in Digital Micrograph.

## Solid-state nuclear magnetic resonance (NMR) characterization

Solid-state NMR measurements were performed on a Bruker Ascend 500 spectrometer ($B_0 = 11.7$ T) with a NEO console operating at a $^6$Li resonance frequency of 73.578 MHz and a $^7$Li resonance frequency of 194.317 MHz. Chemical shifts were referenced with respect to a 1.0 M LiCl solution. A Bruker two-channel magic-angle spinning (MAS) 4 mm and 1.9 mm goniometer probe was used for $^7$Li and $^6$Li measurements, respectively. The MAS frequency was 30 kHz and the probe temperature was controlled at different temperatures. $T_1$ measurements were performed using the saturation recovery experiment at temperatures in the range of 0 to +160 °C. Single-pulse $^6$Li experiments were performed with π/2 pulse lengths of 2.85 μs and the MAS frequency was 30 kHz.

## First-principles calculations

Density functional theory (DFT) calculations were performed with the Vienna *ab* initio simulation package (VASP)[44] adopting the projector augmented wave (PAW)[45,46] approach. The exchange-correlation function was taken the form of the parameterization of Perdew, Burke, and Ernzerhof (PBE)[47] under the generalized gradient approximation (GGA). All the calculations are spin polarized. A van der Waals density functional (vdW-DF, optB86b-vdW) was employed to address the interaction in layered halide structures[48]. The cut-off energy for plane-wave basis was set to 520 eV and a $k$-mesh with the density of one point per -0.03 Å$^{-3}$ is generated using the Gamma-Centered method to ensure the precision of the calculated total energy. The convergence criterion for the electronic self-consistency iteration was $10^{-7}$ eV. During the structural optimization of the models, the forces felt by each of the atoms are well converged below 0.01 eV Å$^{-1}$.

## Surface energy calculations

The surface energy of each orientation can be determined by the following equations[49]:

$$\gamma = \frac{1}{2A}\left(E_{slab} - N \times E_{bulk} - \sum_i N_i \times \mu_i\right) \tag{2}$$

where $\gamma$ is the surface energy of one specific orientation, $A$ and $E_{slab}$ are the cross-sectional area and the total energy of the slab, respectively. The $E_{bulk}$ is the energy of the $Li_3MeCl_6$ primary cell and $N$ is the number of primary cells in the slab. The $\mu_i$ and $N_i$ are the chemical potentials of atoms and the numbers of reduced or added atoms, respectively.

To calculate the surface energy of off-stoichiometry compounds, the chemical potentials of elements ($\mu(Li)$, $\mu(In)$ and $\mu(Cl)$) can be calculated using the grand canonical linear programming (GCLP) method[50] as implemented in the open quantum materials database (OQMD)[46,51]. Briefly, the chemical potentials are determined by mapping the free energy minimization in the Li-In-Cl ternary phase space to a linear algebra problem. Here, the $InCl_2$-$Li_3InCl_6$-LiCl ternary phase region is chosen to calculate the chemical potentials. By solving the equations below, the chemical potentials can be calculated:

$$E(InCl_2) = \mu(In) + 2\mu(Cl) \tag{3}$$

$$E(Li_3InCl_6) = 3\mu(Li) + \mu(In) + 6\mu(Cl) \tag{4}$$

$$E(LiCl) = \mu(Li) + \mu(Cl) \tag{5}$$

where $E(InCl_2)$, $E(Li_3InCl_6)$ and $E(LiCl)$ refer to the total energies of $InCl_2$, $Li_3InCl_6$ and LiCl, respectively.

## Wulff structure construction

According to the Wulff theory[52], the equilibrium shape of crystals contains specific planes to achieve minimum surface energy:

$$\int \gamma_i dA_i = minimum \tag{6}$$

where $A_i$ and $\gamma_i$ refer to the area and surface energy of one specific orientation.

For orientations with surface energy $\gamma$ and radius $R$ (distance from the center of crystal to the interface at the corresponding orientation), the $\gamma$ and $R$ follow the equation below:

$$\frac{\gamma_1}{R_1} = \frac{\gamma_2}{R_2} = \ldots = Constant \tag{7}$$

Therefore, the approximate shape of crystals could be predicted by calculating their surface energies of different orientations and determining the radii correspondingly.

## Data availability
The data that support the findings within this paper are available from the corresponding author on request.

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

## Acknowledgements

The authors acknowledge the financial support by the Netherlands Organization for Scientific Research (NWO) grant 16122 (M.W.). National Natural Science Foundation of China grant 51991344 (X.B.), Chinese Academy of Sciences grant XDB33000000 (X.B.). Special Fund Project for Strategic Emerging Industry Development of Shenzhen grant 20170428145209110 (B.L.), Local Innovative and Research Teams Project of Guangdong Pearl River Talents Program grant 2017BT01N111(B.L.).

## Author contributions

Q.W., C.Z., and M.W. conceived the concept. Data collection and analysis were conducted by Q.W., Y.Z., and C.Z., who carried out the synthesis, material characterization, and electrochemical measurements. H.G. collected the NPD data and interpreted the data by Q.W. J.W. and X.B. performed the TEM measurement. S. Gong. and Z.Y. conducted DFT calculations and analyzed the surface energy. Q.W., and S. Ganapathy conducted the solid-state NMR measurements. F.W. and B.L. carried out the SEM measurements. All authors participated in discussing the results. Q.W., C.Z., X.W., Z.Y., J.J., and M.W. discussed the mechanism. Q.W., C.Z., M.W., and J.J. prepared this manuscript with inputs from all authors.

## Competing interests

The authors declare no competing interests.
