## [Peer Review File · Nature Communications]

REVIEWER COMMENTS

Reviewer #1 (Remarks to the Author):

The authors have described a promising method to identify and design solid halide electrolytes having complex compositions and other relevant attributes like high and isotropic ionic conductivity, isometric morphology, thereby demonstrating improved performances when integrated in an electrochemical cell. Further, identifying such complex compositions can be done in a manner that is computationally less expensive and less complicated than conventional ab initio DFT methods.

Specifically, the authors introduce a parameter called the "ionic potential", that takes into account effective charge on an ionic species as well as the ionic radii. Such a descriptor is able to better quantify interactions in inorganic crystalline compounds, as opposed to relying on ionic radii alone. Moreover, introduction of multiple elements in the composition to study high-entropy (HE) materials can be accommodated comfortably using such a method.

Overall, the manuscript is well-written, detailed, and discusses a novel method supported adequately by data, and I would recommend publication of the same with minor edits and clarifications (listed below) that I feel would be useful to the reader.

1) The language of the manuscript on the whole is scientific and consistent throughout. However, the authors should give it a more careful read to weed out sporadic typos and grammatical errors (such as "blow" instead of "below" - line 169, etc.) The section on "Electrochemical performance in ASSBs" appeared to be inconsistent with the language of the rest of the manuscript, and at times hard to follow. Simplifying the language and using shorter sentences will help communicate ideas better.

2) The authors have defined the effective ionic potential ratio between the Me and X ions as $\Phi_{\text{Me}}/\Phi_{\text{X}}$. It is still unclear as to the rationale behind the choice of numerator & denominator in this ratio. In particular, the ratio of metal to anion radius $R_{\text{Me}}/R_{\text{X}}$ has been suggested in literature to relate phase stability. But if the term Φ is inversely rehashed to R, choosing such a ratio seems quite arbitrary.

3) In the section on designing layered Li-ion halide HE-SEs, a comment is made on increasing the effective ionic potential ratio above a certain threshold. Since the ionic potential is a function not only of charge number but also of the ionic radii, the introduction of more Mes with a higher charge number leading to lower Li-ion conductivity can also be attributed to larger ionic radii. Further, in this

context, it is unclear how optimization of number of vacancies is relevant in the discussion of design in the compositional space.

4) In the same section, the authors set additional criteria for realising high Li-ion conductivity. While use of Cl is understandable owing to superior oxidative stability, choice of Li stoichiometric coefficient to be 2.8 still seems arbitrary & not adequately justified here. How is the average Li-ion potential estimated to be 36.84 nm⁻¹ from the phase map?

5) Later, in the methodology section, authors compare size of Li₃InCl₆ and HE-SE particles using Fig 2e and Suppl figs 18,19. However, SEM images in Suppl figs 18,19 have different scale bars & hence the difference in sizes is not obvious from these images, as claimed.

6) Broadening of resonance peak in MAS-NMR data to study local Li-ion environment suggest disordered distribution in Li environments combined with a change in Li-ion site occupancies. If this be the case, then how can Li-ion diffusion through the HE-SE material occur through a more homogeneous jump process, especially in a layered structure where diffusion is further constrained.

7) The authors also attribute occurrence of more three-dimensional diffusion to observed stacking faults & high cation site disorders. Is there evidence of these defects measured by the authors to support such a claim?

8) While discussing morphological characteristics, it has been indicated that Li₃InCl₆ and the HE-SE were prepared by the same synthesis method, and subsequently show different morphologies. Considering the fact that the HE-SE is significantly different from Li₃InCl₆ in composition, and that morphology, material composition & synthesis method can be highly interdependent, is it fair to make such a comparison of the morphologies for such a complex material system?

9) Can the authors also comment on the expense in terms of computational time required to estimate/map the ionic potential vs compositional space compared to conventional DFT/ab initio methods?

10) Towards the end, the authors highlight the limitations of this approach in the context of dynamic processes. Elaborating on this a little further will help readers better contextualise the capabilities & limitations of this method. Also, can this treatment be extended to other class of SEs like sulphides, oxides, etc.?

In conclusion, I would like to commend the authors on presenting a very thorough study conceptualising a unique method to design & identify a compositional space for high entropy halide SEs having high and isotropic ionic conductivity and isometric morphology.

Reviewer #2 (Remarks to the Author):

This manuscript reports complex multi-component halide solid electrolytes designed by “ionic potential”. These materials exhibited an ionic conductivity of around 2 mS/cm in a C2/m layered structure with isometric morphology. The theme of this research is interesting and well-timed. However, I have some concerns regarding the conduction mechanism of materials and other weaknesses found in this paper. Overall, this work should be published in Nature Communications, subject to the following comments and questions:

- In Figure 1b, the authors suggested the phase map which is well categorized the structure of Li₃MX₆ type of halide SEs. It is needed to discuss whether the concept of ionic potential applies to Na halide SEs as well. Have they tried?
- As the author mentioned, optimization of the number of charge carriers is important to increase ionic conductivity. Therefore, it is recommended to synthesize and analyze HE-SEs without employing aliovalent substitution.
- The conduction pathway of the C2/m layered structure is known as the 3D Li⁺ conduction pathway containing intra-layer Li⁺ migration in the ab-plane at Li slab and the interlayer Li⁺ migration along the c-axis. The author claimed that based on ⁷Li SLR analysis, HE-SEs show more isometric Li⁺ conduction than Li₃InCl₆. Is it possible that disorder resulting from the coexistence of multiple elements within a single lattice site could lead to a symmetric spin-lattice relaxation rate?
- Calculate the activation energy from the ⁷Li SLR NMR and compare it with the results of EIS.
- While ⁷Li NMR of the Li₃InCl₆ show single symmetric peak at 273.15K, ⁷Li NMR of the HE-SE show asymmetric peak at 273.15K. Discuss this asymmetric peak of the HE-SE.
- Assessment of the electrochemical stability window for this HE-SE by CV experiment is desired.
- Does the isometric morphology of HE-SEs assist in regulating the volume change of NCM during the cycle?

Minor Comments:

- The vendor of the materials utilized in the synthesis of halide solid electrolytes should be reported.

- Please revise the solid electrolyte to SE (Line 98, 358)
- Please double-check the Supplementary Tables 12, 15

Reviewer #3 (Remarks to the Author):

This manuscript demonstrates that the ratio of ion potential, charge number and ion radius can capture the key interactions within halide materials, and design a novel chloride solid electrolyte. Although the performance is quite promising, it is necessary to address the following comments for the publication on Nat. Comm.

- (1) It is recommended to cite recent papers on halide electrolytes for solid-state batteries.
- (2) At present, the conductivity of the state-of-the-art oxyhalide solid electrolytes has reached 12.4 mS cm⁻¹, what advantages do HE-SEs have over the above solid electrolytes? Could you elaborate on it?
- (3) The HE-SEs inherit the good electrochemical stability of the halide electrolyte, how about the stability of the Li metal anode? The electrochemical stability was studied by linear sweep voltammetry.
- (4) In addition, NCM should be operated at least up to 4.4V or higher, which is a general condition in the industry.
- (5) In Supplementary Fig.40, the conductivity of HE-SEs is significantly better than Li₃InCl₆, but in fact the conductivity of the two is comparable (~2.2 mS cm⁻¹ vs. 2 mS cm⁻¹), could you explain it? (Angew. Chem. 2019, 131, 16579 –16584)
- (6) The author asserts that the HE-SE has a denser packing, however, its cross-sectional image does not display a homogeneous and dense morphology. Could you explain it?
- (7) Please use first principles computation to investigate the Li-ion diffusion, electrochemical stability, and interface stability of HE-SE and elucidated the origin of their high ionic conductivities and good electrochemical stabilities.

(8) Please explain why the small plateaus at the end of charging and the beginning of discharging caused by the initial reversible phase transition of NCM in Supplementary Fig.34 is different from the subsequent cycle.

(9) Please explain why the first three cycles were performed at 0.1C when investigating the cycling stability of the battery in Fig.4e.

Reviewer #1 (Remarks to the Author):

The authors have described a promising method to identify and design solid halide electrolytes having complex compositions and other relevant attributes like high and isotropic ionic conductivity, isometric morphology, thereby demonstrating improved performances when integrated in an electrochemical cell. Further, identifying such complex compositions can be done in a manner that is computationally less expensive and less complicated than conventional ab initio DFT methods.

Specifically, the authors introduce a parameter called the "ionic potential", that takes into account effective charge on an ionic species as well as the ionic radii. Such a descriptor is able to better quantify interactions in inorganic crystalline compounds, as opposed to relying on ionic radii alone. Moreover, introduction of multiple elements in the composition to study high-entropy (HE) materials can be accommodated comfortably using such a method.

Overall, the manuscript is well-written, detailed, and discusses a novel method supported adequately by data, and I would recommend publication of the same with minor edits and clarifications (listed below) that I feel would be useful to the reader.

Reply: We thank the reviewer for the constructive suggestions and assessment of this manuscript.

- 1) The language of the manuscript on the whole is scientific and consistent throughout. However, the authors should give it a more careful read to weed out sporadic typos and grammatical errors (such as "blow" instead of "below" - line 169, etc.) The section on "Electrochemical performance in ASSBs" appeared to be inconsistent with the language of the rest of the manuscript, and at times hard to follow. Simplifying the language and using shorter sentences will help communicate ideas better.

Reply: We are sorry for these typos, and carefully checked and revised the text.

- 2) The authors have defined the effective ionic potential ratio between the Me and X ions as Φ_{Me}/Φ_X . It is still unclear as to the rationale behind the choice of numerator & denominator in this ratio. In particular, the ratio of metal to anion radius R_{Me}/R_X has been suggested in literature to relate phase stability. But if the term Φ is inversely rehashed to R , choosing such a ratio seems quite arbitrary.

Reply: The choice of numerator and denominator of effective ionic potential ratio Φ_{Me}/Φ_X takes into account the in literature suggested ratio of radii, R_{Me}/R_X . However, this ratio alone is not sufficient to distinguish the different structural arrangements. The term 'ionic radius' signifies the size of an ion within an ionic crystal structure, which does not consider the influence of the charge of the ions. In the case of a crystal halide compound, its crystal lattices can be described by ionic Me–X bonds present within the closely packed anion sublattice. The strength of ionic interactions between Me and X ions plays a pivotal role in determining the various structural arrangements. By utilizing the 'ionic potential' (Φ), which serves as an indicator of the charge density at the ion's surface and is calculated as the ratio of the charge number (n) to the ion radius (R), we bring forward a descriptor that is more accurate than the ratio of the radii. This effectively represents a simple descriptor of the ionic interactions within this structural family. We have highlighted this reasoning in the manuscript

- 3) In the section on designing layered Li-ion halide HE-SEs, a comment is made on increasing the effective ionic potential ratio above a certain threshold. Since the ionic potential is a function not only of charge number but also of the ionic radii, the introduction of more Mes with a higher charge number leading to lower Li-ion conductivity can also be attributed to larger ionic radii. Further, in this context, it is unclear how optimization of number of vacancies is relevant in the discussion of design in the compositional space.

Reply: For most cases, the Me ions with a higher charge number used in the halides generally exhibit relatively smaller ionic radii, as demonstrated in Supplementary Table 5. For example, the Zr^{4+} ionic radius (6-coordination) is 0.072 nm, which is smaller than that of Sc^{3+} (0.0745 nm), Y^{3+} (0.09 nm) and In^{3+} (0.08nm). Consequently, the introduction of more Me ions with higher charge numbers results in an increased ionic potential of $\overline{\Phi_{Me}}$, leading to a decreased $d_{(X-Me-X)}$ and the increased $d_{(X-Li-X)}$ distances. This change does not necessarily increase the Li-ion conductivity. An example of the monoclinic Li_2ZrCl_6 ¹ exhibits an ion conductivity of approximately 10^{-6} S cm^{-1} at room temperature, which is lower than that of Li_3InCl_6 . This anomaly can be attributed to the reduced number of charge carriers, resulting in lower ionic conductivity². This has been clarified in the manuscript for clarification.

- 4) In the same section, the authors set additional criteria for realising high Li-ion conductivity. While use of Cl is understandable owing to superior oxidative stability, choice of Li stoichiometric coefficient to be 2.8 still seems arbitrary & not adequately justified here. How is the average Li-ion potential estimated to be 36.84 nm⁻¹ from the phase map?

Reply: There are two factors to consider when choosing a Li stoichiometric coefficient of 2.8. The first consideration is based on previously reported solid electrolytes, such as $Li_{6.5}La_3Zr_{1.5}Ta_{0.5}O_{12}$ ³, where the parent composition is $Li_7La_3Zr_2O_{12}$. In this case, 0.5 mole fraction of Li^+ vacancies are introduced on a total of 7 moles. Similarly, in the context of Li_3MeX_6 , we introduce 0.2 moles vacancies out of the 3 moles of Li^+ . The second reason for choosing 2.8 involves achieving an equal proportion of five Me elements and vacancies to maximize the configurational entropy. We have revised the related text to make these motivations much clear.

The average Li-ion potential of 36.84 nm⁻¹ is calculated based on the 2.8 mol Li^+ in $Li_{2.8}MeX_6$, which refers to $\overline{\Phi_{Li}} = \frac{y}{R_{Li}}$ and y is the stoichiometric coefficient, which equals to $2.8 \times 1/0.076 = 36.8421$ nm⁻¹.

- 5) Later, in the methodology section, authors compare size of Li_3InCl_6 and HE-SE particles using Fig 2e and Suppl figs 18,19. However, SEM images in Suppl figs 18,19 have different scale bars & hence the difference in sizes is not obvious from these images, as claimed.

Reply: We thank the reviewer for pointing this out. The SEM images of the two Li_3InCl_6 and HE-SE materials with the same scale bars are already shown in Fig. 3a and 3b. However, to make it consistent, the Supplementary Figs. 18 and 19 are also changed into the same scale bars.

- 6) Broadening of resonance peak in MAS-NMR data to study local Li-ion environment suggest

disordered distribution in Li environments combined with a change in Li-ion site occupancies. If this be the case, then how can Li-ion diffusion through the HE-SE material occur through a more homogeneous jump process, especially in a layered structure where diffusion is further constrained.

Reply: For the Li_3InCl_6 material, the temperature dependence of the spin-lattice relaxation rate reflects two distinguishable peaks (Supplementary Fig. 25), suggesting the presence of two Li-ion jump processes that differ in their average jump frequency. Considering the two-dimensional layered structure, this has been suggested to be the result of the relatively fast intralayer diffusion within the LiCl_2 slabs and relatively sluggish diffusion across the InCl_2 slabs, thus between two LiCl_2 slabs⁴. In contrast, a symmetric spin-lattice relaxation rate is observed for the HE-SE material, showing that Li-ion diffusion within the LiCl_2 slabs, as well as diffusion across the MeCl_2 slabs between two LiCl_2 slabs have a similar average jump frequency, which reflects a more homogeneous jump process⁵⁻⁷. We have revised the related text to make this point clearer.

- 7) The authors also attribute occurrence of more three-dimensional diffusion to observed stacking faults & high cation site disorders. Is there evidence of these defects measured by the authors to support such a claim?

Reply: Yes. A stacking fault is a planar defect that can occur in crystalline materials, which is commonly found in close-packed crystal structures, like in layered oxide cathodes^{8,9} and halide SEs¹⁰, reflected by broadened super-lattice peaks. Here, the broadened superstructure peaks between 5.5° to 10° in the XRD data agree well with this, which suggests that the HE-SE has a higher concentration of stacking faults as shown in Fig. 2d. Additionally, increased cation-site disorder is indicated by the combined Rietveld refinement of the NPD and XRD data of the HE-SE, where a partially disordered arrangement of Li and Me ions in the MeX_2 slabs is observed with an increased fraction of the Li residing in the Me (4g) site (Supplementary table 12). Further, ^6Li MAS-NMR measurements show a considerably broader resonance of the HE-SE (FWHM approximately 1.65 ppm) compared to that of Li_3InCl_6 (FWHM approximately 0.1 ppm) also indicating disorder in the Li^+ environments. We have revised the related text to make this point much clear.

- 8) While discussing morphological characteristics, it has been indicated that Li_3InCl_6 and the HE-SE were prepared by the same synthesis method, and subsequently show different morphologies. Considering the fact that the HE-SE is significantly different from Li_3InCl_6 in composition, and that morphology, material composition & synthesis method can be highly interdependent, is it fair to make such a comparison of the morphologies for such a complex material system?

Reply: We agree that the morphology, material composition and synthesis method are highly interdependent. In this case the two materials are prepared using the same synthesis method, where the simple compositions always show platelet, strongly anisotropic particle shapes. This is consistent with reported halides such as Li_3InCl_6 ¹¹, Li_3ScCl_6 ¹², and Li_3YBr_6 ¹³. Even though they both have layered structure and prepared in the same way, all complex, high entropy compositions result in an isotropic particle shape, therefore we think it is justified to make this comparison. Additionally, the surface energy simulations support that the origin of the difference in crystallite

shape is the surface energy, which is intrinsic in nature.

- 9) Can the authors also comment on the expense in terms of computational time required to estimate/map the ionic potential vs compositional space compared to conventional DFT/ab initio methods?

Reply: In the context of conventional DFT ab initio methods, several key considerations come into play: including computational scaling, resource requirements, accuracy vs. efficiency trade-off, parallelization and software, and so on. Clearly, the huge configurational space of the presented complex halide compositions introduces relatively long computational times. It is difficult to quantify this as it depends on the exact composition and computational cluster. Our experience is that more simple disordered systems can take already weeks to months to scan through configurational space to identify the lowest energy configuration. On the contrary, the ionic potential is a straightforward calculation based on the composition which is easily done by hand on paper.

- 10) Towards the end, the authors highlight the limitations of this approach in the context of dynamic processes. Elaborating on this a little further will help readers better contextualise the capabilities & limitations of this method. Also, can this treatment be extended to other class of SEs like sulphides, oxides, etc.?

Reply: This for instance refers to amorphous phases or metastable structures resulting from mechanical milling or particular synthesis conditions. For example, in addition to the stable hexagonal (*P-3m1*) Li_3YCl_6 and (*C2/m*) Li_2ZrCl_6 phases, the metastable (*Pnma*) Li_3YCl_6 ¹⁴, and (*P-3m1*) Li_2ZrCl_6 ¹⁵ also exist, which can be directly prepared by mechanical milling. Here the ionic potential method does not provide a sensible guideline, because one composition has only one calculated value for the ionic potential ratio. This has been added to the manuscript for clarification. We are currently investigating if the ionic potential method can be extended to other SE classes, where we anticipate it will be most successful in the more ionic bond type crystals.

In conclusion, I would like to commend the authors on presenting a very thorough study conceptualising a unique method to design & identify a compositional space for high entropy halide SEs having high and isotropic ionic conductivity and isometric morphology.

Reply: We thank the reviewer again for this high evaluation of our work.

Reviewer #2 (Remarks to the Author):

This manuscript reports complex multi-component halide solid electrolytes designed by “Ionic potential”. These materials exhibited an ionic conductivity of around 2 mS/cm in a C2/m layered structure with isometric morphology. The theme of this research is interesting and well-timed. However, I have some concerns regarding the conduction mechanism of materials and other weaknesses found in this paper. Overall, this work should be published in Nature Communications, subject to the following comments and questions:

Reply: We sincerely thank the reviewer for this recommendation and constructive suggestions.

- In Figure 1b, the authors suggested the phase map which is well categorized the structure of Li3MX6 type of halide SEs. It is needed to discuss whether the concept of ionic potential applies to Na halide SEs as well. Have they tried?

Reply: We thank the reviewer for this suggestion. We indeed considered applying the ionic potential method to Na halide SEs; however, reported Na halides do not share the three structures with Li halides. For example, Na₃AlF₆ is the *P*_{21/n} phase¹⁶, Na₃ScF₆ is the *P*_{21/n} phase¹⁷, Na₃InF₆ is the *P*_{21/c} phase¹⁸, Na₃MeCl₆ (Me= Y, Sc and Dy-Lu) is the *P*_{21/n} phase, Na₃MeCl₆ (Me= Eu, Gd, Tb) is the *R*-3 phase¹⁹, Na₃MeBr₆ (Me= Sm-Gd) is the *R*-3 phase²⁰, Na₃MeBr₆ (Me=Y, Gd-Lu) is the *P*_{21/n} phase^{19,20}, Na₂ZrCl₆ is *P*-3m₁ phase²¹, Na_{3-x}Y_{1-x}Zr_xCl₆ (x = 0, 0.25, 0.5, 0.75) is the *P*_{21/n} phase²², Na₃InCl₆ is the *P*-3_{1c} phase²³. We are investigating the Na halides phase maps, where we anticipate that an adapted or new structural classification may be required, which is outside the scope of this work and the work is in progress.

- As the author mentioned, optimization of the number of charge carriers is important to increase ionic conductivity. Therefore, it is recommended to synthesize and analyze HE-SEs without employing aliovalent substitution.

Reply: We agree this is important. The HE-SEs without aliovalent substitution, for example of the Li₃In_{0.2}Sc_{0.2}Yb_{0.2}Lu_{0.2}Y_{0.2}Cl₆ composition has also been synthesized and studied within this work. As prepared and tested under the same condition, it has an ionic conductivity up to about 1.76 mS cm⁻¹, showing only a slightly lower conductivity compared to the high entropy material with aliovalent substitution. This has been added into the test.

- The conduction pathway of the C2/m layered structure is known as the 3D Li⁺ conduction pathway containing intra-layer Li⁺ migration in the ab-plane at Li slab and the interlayer Li⁺ migration along the c-axis. The author claimed that based on 7Li SLR analysis, HE-SEs show more isometric Li⁺ conduction than Li₃InCl₆. Is it possible that disorder resulting from the coexistence of multiple elements within a single lattice site could lead to a symmetric spin-lattice relaxation rate?

Reply: This is indeed what the results suggest, where the relevant text can be found on page 7-8 in the manuscript. The symmetric spin-lattice relaxation rate suggests more isotropic jump diffusion which may be attributed to the observed stacking faults, a disordered Li-ion distribution and increased cation disorder resulting from the presence of multiple Me ions on the same sublattice.

We have revised the related text to make this point clear.

- Calculate the activation energy from the ^7Li SLR NMR and compare it with the results of EIS.

Reply: The activation energies from the ^7Li SLR NMR of the two SEs are 0.075 ± 0.021 eV and 0.122 ± 0.011 eV for the HE-SE and Li_3InCl_6 at the low-temperature regions, respectively. The results indicate the HE-SE shows improved local Li-ion diffusion and lower Li-ion diffusion energy barrier, which are in good agreement with impedance tests. But it is important to note that the values of the activation energies from NMR are attributed to local Li-ion motion, not taking into account the contributions from long range effects such as grain boundaries, thus representing a combination of the short-range, local Li-ion motional processes⁵⁻⁷. This discussion has been added into the supplementary information.

- While ^7Li NMR of the Li_3InCl_6 show single symmetric peak at 273.15K, ^7Li NMR of the HE-SE show asymmetric peak at 273.15K. Discuss this asymmetric peak of the HE-SE.

Reply: Since HE-SE is observed to have abundant stacking faults and a highly disordered distribution of Li and Me ion environments, especially including an increased Li occupancy on the Me (4g) site, this is held responsible for the observed chemical shift anisotropy of Li^{+24} . At a temperature of 273.15 K, the asymmetric peak of the static ^7Li NMR pattern of the HE-SE suggest that the motion of the Li^+ across the different sites does not average out completely. As the temperature increases, the spectra become narrower and more symmetric, reflecting that mobile Li-ions increasingly average out the dipolar interactions, thereby observing a more average environment^{6,25}. This discussion has been added into the supplementary information of the Supplementary Figure 27.

- Assessment of the electrochemical stability window for this HE-SE by CV experiment is desired.

Reply: The CV experiments of the two SEs are shown in the Fig. R1. Compared to the Li_3InCl_6 SE, this prepared HE-SEs shows an improved oxidation and reduction stability window.

Fig. R1. CV experiments of the SEs. The measurements were carried out at a scan rate of 0.2 mV s^{-1} from -0.1 to 5 V vs. Li/Li^+ in the cell configuration of $\text{Li|LPSC-halide|Stainless-steel}$ using

different halide SEs.

- Does the isometric morphology of HE-SEs assist in regulating the volume change of NCM during the cycle?

Reply: We are currently uncertain about this. Our ongoing focus also includes investigating the phase transition of the cathode in solid-state batteries, particularly when utilizing the improved HE-SEs. However, we are still facing challenges due to the complexities associated with operando XRD measurements.

Minor Comments:

- The vendor of the materials utilized in the synthesis of halide solid electrolytes should be reported.
- Please revise the solid electrolyte to SE (Line 98, 358)
- Please double-check the Supplementary Tables 12, 15

Reply: Thank you very much. These comments have been addressed.

Reviewer #3 (Remarks to the Author):

This manuscript demonstrates that the ratio of ion potential, charge number and ion radius can capture the key interactions within halide materials, and design a novel chloride solid electrolyte. Although the performance is quite promising, it is necessary to address the following comments for the publication on Nat. Comm.

Reply: We thank the reviewer for positive recommendation of our work for publication.

(1) It is recommended to cite recent papers on halide electrolytes for solid-state batteries.

Reply: Several recent papers related to halide electrolytes have been added as below. If there are any relevant articles that are missed here, we welcome suggestions.

- Tanaka et al., *Angew. Chem. Int. Ed.* 2023, 62, e202217581.
- Kwak et al., *Nat Commun* 14, 2459 (2023).
- Hu et al., *Nat Commun* 14, 3807 (2023).
- Zhang et al., *Nat Commun* 14, 3780 (2023)
- Ishiguro et al., *Chemistry Letters* 2023 52:4, 237-241.
- Li et al., *J. Am. Chem. Soc.* 2023, 145, 21, 11701-11709.
- Yin et al., *Nature* 616, 77-83 (2023).

(2) At present, the conductivity of the state-of-the-art oxyhalide solid electrolytes has reached 12.4 mS cm⁻¹, what advantages do HE-SEs have over the above solid electrolytes? Could you elaborate on it?

Reply: We are happy to see that these oxyhalide solid electrolytes present such high ionic conductivity, which are promising candidates for ASSBs^{26,27}. In general, we argue that up to date there is no single SE that addresses all demands (high conductivity, high stability, facile production, low moisture/air sensitivity, low critical material dependence, low cost and CO₂ footprint). The aim of the present work is not to argue that the present HE-SEs are better than other classes of SEs, nor to provide a comprehensive comparison. In our view the development of SEs is an ongoing research area where there is value in the discovery of new materials and concepts, aiming to improve the combination of functional properties. Here we introduce a design guideline for complex halides, which is shown to improve functional SE properties. Focusing on the comparison with oxyhalides, as stated in the reference^{26,27}, the oxyhalide solid electrolytes show indeed high ionic conductivity around 10 mS cm⁻¹, consisting Me ions with higher charge numbers, such as Nb⁵⁺ and Ta⁵⁺. Consequently, these SEs are found to exhibit a relatively lower reduction stability with an onset voltage of around 2.4 to 3.0 V, suggesting potential space for further improvement. In addition, based on our understanding, these oxyhalides used as electrolytes present a glass/amorphous phase, which usually requires more energy intensive synthesis methods, up to 200 h using mechanochemical milling at a speed of 500 rpm min⁻¹²⁶. Meanwhile, they could be limited in their application by the lower abundance of Ta/Nb elements in the earth's crust (approximately 2/20 ppm) and the high price of the main raw materials. A potential advantage of the presented HE-SEs is that these utilize Me ions with lower charge numbers of 3+/4+, which leads to a higher reduction stability of around 1.0 V as shown in Fig. R1 shown in comment #3. Additionally, they can be prepared using

a shorter milling time of 15 h at a lower speed 350 rpm min⁻¹. Also, the use of multiple Me elements could effectively balance the cost-effectiveness. Perhaps interesting future work could be to investigate if the high entropy approach provides opportunities to further develop oxyhalide materials.

(3) The HE-SEs inherit the good electrochemical stability of the halide electrolyte, how about the stability of the Li metal anode? The electrochemical stability was studied by linear sweep voltammetry.

Reply: The CV experiments of the two SEs are shown in the Fig. R1. This prepared HE-SEs shows an improved oxidation and reduction stability window, compared to the Li₃InCl₆ SE. However, the reduction stability of around 1.0 V is not compatible with Li metal.

Fig. R1. CV experiments of the SEs. The measurements were carried out at a scan rate of 0.2 mV s⁻¹ from -0.1 to 5 V vs. Li/Li⁺ in the cell configuration of Li|LPSC-halide|Stainless-steel using different halide SEs.

(4) In addition, NCM should be operated at least up to 4.4V or higher, which is a general condition in the industry.

Reply: A higher cut-off voltage would indeed increase the capacity, and we agree this is relevant. In the context of this work aiming for introducing guided high entropy halide SE design, we have not focussed on studying the electrochemical parameters of ASSBs. Instead, we adopted testing parameters that are commonly employed in literature, even aligned with recent papers on halide electrolytes²⁶⁻³⁵. Exploring the NCM cathode to a higher voltage will be part of follow up work aiming for optimized cycling performances.

(5) In Supplementary Fig.40, the conductivity of HE-SEs is significantly better than Li₃InCl₆, but in fact the conductivity of the two is comparable (~2.2 mS cm⁻¹ vs. 2 mS cm⁻¹), could you explain it? (Angew. Chem. 2019, 131, 16579 –16584)

Reply: The work pointed out by this reviewer is from Sun's group published on 30th Sept. 2019, and the first author of this work also reports another work on Li_3InCl_6 (Energy Environ. Sci., 2019,12, 2665-2671) published on 28th Aug. 2019. The Li_3InCl_6 material in these two works were prepared using a very different synthesis methods, a water-mediated synthesis in Angew. Chem. (which is specifically suitable for Li_3InCl_6) and a ball milling following by a solid-state reaction in Energy Environ. Sci. Using the solid-state reaction, the Li_3InCl_6 material shows ionic conductivities from 1.03 to 1.31 mS cm^{-1} depending on different annealing parameters, which is consistent with our study. It is very important and relevant to investigate variations in conductivity between materials synthesized using different methods. Here we aimed to evaluate the impact of high entropy compositions, and guidance to the design of these, and therefore using the more common synthesis method for halide materials which is ball milling. This has been highlighted in the manuscript.

(6) The author asserts that the HE-SE has a denser packing, however, its cross-sectional image does not display a homogeneous and dense morphology. Could you explain it?

Reply: The cross-sectional images of the SEs can be found in Supplementary Figure 29 and 30, also Fig. R2. When comparing the HE-SE to Li_3InCl_6 SE, it is shown that the HE-SE, with its smaller and more spherical particles, shows reduced grain boundary formation. This suggests a relatively denser packing in the HE-SE. However, we should note the cross-sectional samples were prepared by manually breaking electrolyte pellets, which could potentially present irregularities or bumps.

Fig. R2. Cross-section view of the SE pellets.

(7) Please use first principles computation to investigate the Li-ion diffusion, electrochemical stability, and interface stability of HE-SE and elucidated the origin of their high ionic conductivities and good electrochemical stabilities.

Reply: We thank the reviewer for this suggestion. Using the density functional theory simulations to assess the properties of HE-SEs is also our interest. However, it is important to acknowledge that the increased complexity with multiple Me components and multiple Li/Me/Cl sites leads to a significantly expanded configurational space for optimization, making it very challenging to handle with the current computational resources. This is also the reason why we developed the ionic potential method to predict the structures of HE halide.

(8) Please explain why the small plateaus at the end of charging and the beginning of discharging caused by the initial reversible phase transition of NCM in Supplementary Fig.34 is different from

the subsequent cycle.

Reply: This could be related to the possible phase transition of the NCM cathodes or/and the formation of cathode–electrolyte interphase^{36,37}.

(9) Please explain why the first three cycles were performed at 0.1C when investigating the cycling stability of the battery in Fig.4e.

Reply: The first three cycles performed at 0.1C was used as the pre-cycling to activated the cathode materials. This has been added in the manuscript for clarification.

Reference

- 1 Kwak, H. *et al.* Li⁺ conduction in aliovalent-substituted monoclinic Li₂ZrCl₆ for all-solid-state batteries: Li_{2+x}Zr_{1-x}M_xCl₆ (M = In, Sc). *Chemical Engineering Journal* **437**, 135413 (2022).
- 2 Mouta, R., Melo, M. Á. B., Diniz, E. M. & Paschoal, C. W. A. Concentration of Charge Carriers, Migration, and Stability in Li₃OCl Solid Electrolytes. *Chemistry of Materials* **26**, 7137-7144 (2014).
- 3 Liu, J. *et al.* The Interface between Li_{6.5}La₃Zr_{1.5}Ta_{0.5}O₁₂ and Liquid Electrolyte. *Joule* **4**, 101-108 (2020).
- 4 Helm, B. *et al.* Exploring Aliovalent Substitutions in the Lithium Halide Superionic Conductor Li_{3-x}In_{1-x}Zr_xCl₆ (0 ≤ x ≤ 0.5). *Chemistry of Materials* **33**, 4773-4782 (2021).
- 5 Epp, V., Gün, Ö., Deiseroth, H.-J. & Wilkening, M. Highly Mobile Ions: Low-Temperature NMR Directly Probes Extremely Fast Li⁺ Hopping in Argyrodite-Type Li₆PS₅Br. *The Journal of Physical Chemistry Letters* **4**, 2118-2123 (2013).
- 6 Yu, C. *et al.* Unravelling Li-Ion Transport from Picoseconds to Seconds: Bulk versus Interfaces in an Argyrodite Li₆PS₅Cl–Li₂S All-Solid-State Li-Ion Battery. *Journal of the American Chemical Society* **138**, 11192-11201 (2016).
- 7 Ganapathy, S., Yu, C., van Eck, E. R. H. & Wagemaker, M. Peeking across Grain Boundaries in a Solid-State Ionic Conductor. *ACS Energy Letters* **4**, 1092-1097 (2019).
- 8 Boulineau, A., Croguennec, L., Delmas, C. & Weill, F. Reinvestigation of Li₂MnO₃ Structure: Electron Diffraction and High Resolution TEM. *Chemistry of Materials* **21**, 4216-4222 (2009).
- 9 Bréger, J. *et al.* High-resolution X-ray diffraction, DIFFaX, NMR and first principles study of disorder in the Li₂MnO₃–Li[Ni_{1/2}Mn_{1/2}]O₂ solid solution. *Journal of Solid State Chemistry* **178**, 2575-2585 (2005).
- 10 Sebti, E. *et al.* Stacking Faults Assist Lithium-Ion Conduction in a Halide-Based Superionic Conductor. *Journal of the American Chemical Society* **144**, 5795-5811 (2022).
- 11 Wang, S. *et al.* Lithium Chlorides and Bromides as Promising Solid-State Chemistries for Fast Ion Conductors with Good Electrochemical Stability. *Angewandte Chemie International Edition* **58**, 8039-8043 (2019).
- 12 Liang, J. *et al.* Site-Occupation-Tuned Superionic Li_xScCl_{3+x}Halide Solid Electrolytes for All-Solid-State Batteries. *Journal of the American Chemical Society* **142**, 7012-7022 (2020).

- 13 Yu, T. *et al.* Superionic Fluorinated Halide Solid Electrolytes for Highly Stable Li-Metal in All-Solid-State Li Batteries. *Advanced Energy Materials* **11**, 2101915 (2021).
- 14 Bohnsack, A. *et al.* Ternäre Halogenide vom Typ A₃MX₆. VI [1]. Ternäre Chloride der Selten-Erd-Elemente mit Lithium, Li₃MCl₆ (M □ Tb□Lu, Y, Sc): Synthese, Kristallstrukturen und Ionenbewegung. *Zeitschrift für anorganische und allgemeine Chemie* **623**, 1067-1073 (1997).
- 15 Wang, K. *et al.* A cost-effective and humidity-tolerant chloride solid electrolyte for lithium batteries. *Nature Communications* **12**, 4410 (2021).
- 16 Hawthorne, F. C. & Ferguson, R. B. Refinement of the crystal structure of cryolite. *The Canadian Mineralogist* **13**, 377-382 (1975).
- 17 Bohnsack, A. & Meyer, G. Ternäre Halogenide vom Typ A₃MX₆. IV. [1]. Ternäre Halogenide des Scandiums mit Natrium, Na₃ScX₆ (X = F, Cl, Br): Synthese, Strukturen, Ionenleitfähigkeit. *Zeitschrift für anorganische und allgemeine Chemie* **622**, 173-178 (1996).
- 18 Cros, C., Feurer, R., Pouchard, M. & Hagenmuller, P. Les bronzes fluores de vanadium. *Materials Research Bulletin* **10**, 383-391 (1975).
- 19 Meyer, G., Peter Ax, S., Schleid, T. & Irmeler, M. The chlorides Na₃MCl₆ (M □ Eu□Lu, Y, Sc): Synthesis, crystal structures, and thermal behaviour. *Zeitschrift für anorganische und allgemeine Chemie* **554**, 25-33 (1987).
- 20 Wickleder, M. S. & Meyer, G. Ternäre Halogenide vom Typ A₃MX₆. III [1, 2]. Synthese, Strukturen und Ionenleitfähigkeit der Halogenide Na₃MX₆ (X = Cl, Br). *Zeitschrift für anorganische und allgemeine Chemie* **621**, 457-463 (1995).
- 21 Schlem, R., Banik, A., Eckardt, M., Zobel, M. & Zeier, W. G. Na_{3-x}Er_{1-x}Zr_xCl₆—A Halide-Based Fast Sodium-Ion Conductor with Vacancy-Driven Ionic Transport. *ACS Applied Energy Materials* **3**, 10164-10173 (2020).
- 22 Wu, E. A. *et al.* A stable cathode-solid electrolyte composite for high-voltage, long-cycle-life solid-state sodium-ion batteries. *Nature Communications* **12**, 1256 (2021).
- 23 Yamada, K., Kumano, K. & Okuda, T. Conduction path of the sodium ion in Na₃InCl₆ studied by X-ray diffraction and ²³Na and ¹¹⁵In NMR. *Solid State Ionics* **176**, 823-829 (2005).
- 24 van der Maas, E. *et al.* Re-investigating the structure–property relationship of the solid electrolytes Li_{3-x}In_{1-x}Zr_xCl₆ and the impact of In – Zr(IV) substitution. *Journal of Materials Chemistry A* **11**, 4559-4571 (2023).
- 25 Deiseroth, H.-J. *et al.* Li₆PS₅X: A Class of Crystalline Li-Rich Solids With an Unusually High Li⁺ Mobility. *Angewandte Chemie International Edition* **47**, 755-758 (2008).
- 26 Ishiguro, Y., Ueno, K., Nishimura, S., Iida, G. & Igarashib, Y. TaCl₅-glassified Ultrafast Lithium Ion-conductive Halide Electrolytes for High-performance All-solid-state Lithium Batteries. *Chemistry Letters* **52**, 237-241 (2023).
- 27 Tanaka, Y. *et al.* New Oxyhalide Solid Electrolytes with High Lithium Ionic Conductivity >10 mS cm⁻¹ for All-Solid-State Batteries. *Angewandte Chemie International Edition* **62**, e202217581 (2023).
- 28 Asano, T. *et al.* Solid Halide Electrolytes with High Lithium-Ion Conductivity for Application in 4 V Class Bulk-Type All-Solid-State Batteries. *Advanced Materials* **30**, 1803075 (2018).
- 29 Li, X. *et al.* Progress and perspectives on halide lithium conductors for all-solid-state lithium batteries. *Energy & Environmental Science* **13**, 1429-1461 (2020).

- 30 Zhou, L. *et al.* High areal capacity, long cycle life 4 V ceramic all-solid-state Li-ion batteries enabled by chloride solid electrolytes. *Nature Energy* **7**, 83-93 (2022).
- 31 Kochetkov, I. *et al.* Different interfacial reactivity of lithium metal chloride electrolytes with high voltage cathodes determines solid-state battery performance. *Energy & Environmental Science* **15**, 3933-3944 (2022).
- 32 Kwak, H. *et al.* Boosting the interfacial superionic conduction of halide solid electrolytes for all-solid-state batteries. *Nature Communications* **14**, 2459 (2023).
- 33 Hu, L. *et al.* A cost-effective, ionically conductive and compressible oxychloride solid-state electrolyte for stable all-solid-state lithium-based batteries. *Nature Communications* **14**, 3807 (2023).
- 34 Zhang, S. *et al.* A family of oxychloride amorphous solid electrolytes for long-cycling all-solid-state lithium batteries. *Nature Communications* **14**, 3780 (2023).
- 35 Yin, Y.-C. *et al.* A LaCl₃-based lithium superionic conductor compatible with lithium metal. *Nature* **616**, 77-83 (2023).
- 36 Li, H. *et al.* An Unavoidable Challenge for Ni-Rich Positive Electrode Materials for Lithium-Ion Batteries. *Chemistry of Materials* **31**, 7574-7583 (2019).
- 37 Zhang, N. *et al.* Cobalt-Free Nickel-Rich Positive Electrode Materials with a Core-Shell Structure. *Chemistry of Materials* **31**, 10150-10160 (2019).

REVIEWERS' COMMENTS

Reviewer #1 (Remarks to the Author):

I feel that the response presented by the authors to all comments, questions, and clarifications sought by reviewers is satisfactory. The revised manuscript adequately reflects the edits made and sought.

I once again commend the authors on a well-written and thorough manuscript detailing a unique for identifying and designing complex halide-based solid electrolytes for ASSBs, and I would recommend publication of the same.

Reviewer #2 (Remarks to the Author):

The authors have appropriately addressed all the questions and comments. The manuscript can be accepted for the publication.

Reviewer #3 (Remarks to the Author):

The author has explained my confusion very well, and it is recommended to publish on Nat. Commun.